# The Use of Immune Regulation in Treating Head and Neck Squamous Cell Carcinoma (HNSCC)

**DOI:** 10.3390/cells13050413

**Published:** 2024-02-27

**Authors:** Che-Wei Wang, Pulak Kumar Biswas, Atikul Islam, Mu-Kuan Chen, Pin Ju Chueh

**Affiliations:** 1Institute of Biomedical Sciences, National Chung Hsing University, Taichung 40227, Taiwan; 97302@cch.org.tw (C.-W.W.); islammiu555@gmail.com (A.I.); 2Department of Otorhinolaryngology-Head and Neck Surgery, Changhua Christian Hospital, Changhua 50006, Taiwan; 53780@cch.org.tw; 3Institute of Molecular Medicine, National Cheng Kung University, Tainan 70101, Taiwan; pulak.live@gmail.com

**Keywords:** immunotherapy, immune checkpoint inhibitors, head and neck squamous cell carcinoma (HNSCC), oral carcinoma, programmed cell death protein 1, programmed death ligand-1 (PD-1/PD-L1), cytotoxic T-lymphocyte-associated protein 4 (CTLA-4), tumor microenvironment, combination therapy

## Abstract

Immunotherapy has emerged as a promising new treatment modality for head and neck cancer, offering the potential for targeted and effective cancer management. Squamous cell carcinomas pose significant challenges due to their aggressive nature and limited treatment options. Conventional therapies such as surgery, radiation, and chemotherapy often have limited success rates and can have significant side effects. Immunotherapy harnesses the power of the immune system to recognize and eliminate cancer cells, and thus represents a novel approach with the potential to improve patient outcomes. In the management of head and neck squamous cell carcinoma (HNSCC), important contributions are made by immunotherapies, including adaptive cell therapy (ACT) and immune checkpoint inhibitor therapy. In this review, we are focusing on the latter. Immune checkpoint inhibitors target proteins such as programmed cell death protein 1 (PD-1) and cytotoxic T-lymphocyte-associated protein 4 (CTLA-4) to enhance the immune response against cancer cells. The CTLA-4 inhibitors, such as ipilimumab and tremelimumab, have been approved for early-stage clinical trials and have shown promising outcomes in terms of tumor regression and durable responses in patients with advanced HNSCC. Thus, immune checkpoint inhibitor therapy holds promise in overcoming the limitations of conventional therapies. However, further research is needed to optimize treatment regimens, identify predictive biomarkers, and overcome potential resistance mechanisms. With ongoing advancements in immunotherapy, the future holds great potential for transforming the landscape of oral tumor treatment and providing new hope for patients.

## 1. Introduction

Head and neck cancer is a group of malignancies arising in the epithelial tissues of the paranasal sinuses, lips, oral cavity, nasal cavity, pharynx, and larynx; it is the sixth most common malignancy worldwide, accounting for 4% of all cancers [1,2]. The most representative malignancy of this type of cancer is squamous cell carcinoma, naming head and neck squamous cell carcinoma (HNSCC) [3]. HNSCC can be induced by many factors, including long-term alcoholism, poor oral hygiene, excessive sun exposure, betel nut chewing, and cigarette smoking [4,5,6]. In addition, over the last two to three decades, human papillomavirus (HPV) has been associated with the development of oral squamous cell carcinoma [7,8]. Traditional treatment approaches such as surgery, chemotherapy, radiotherapy, and combination therapy are effective in the treatment of HNSCC. The surgical approach is an established gold standard for the initial treatment of HNSCC patients. However, around 94% (N = 219/234) of the patients had a recurrence within 18 months after surgery. Alternatively, radiotherapy and/or chemotherapy are often used to treat HNSCC. Although reducing tumors, those treatments also pose adverse effects [9,10,11]. Despite massive improvements in treatment strategies for HNSCC, the 5-year overall survival rate is 30–65%. Moreover, approximately 10% of HNSCC patients will have a metastatic disease and a high risk of recurrence [12]. Thus, further improvements in the detection and treatment of HNSCC are highly pivotal [1,13]. Among the aforementioned treatment strategies, chemotherapy and/or anticancer drug therapy are progressing rapidly and becoming more diverse. Recent work introduced molecular targeted therapy by immune checkpoint inhibitors (ICIs), which are becoming increasingly important [14]. Immunotherapies such as ICI therapy aim to increase the activity of the immune system to destroy cancer cells. ICIs are widely effective in blocking inhibitory immune checkpoint signaling pathways to reactivate immune responses against cancer. The interaction between programmed cell death protein 1 (PD-1), which is expressed by T-cells, and its ligand PD-L1, which is commonly expressed by tumor cells, results in the suppression of immunological T-cell responses and serves as a mechanism of tumor immune evasion. Anti-PD-1/PD-L1 ICIs can block PD-1/PD-L1-mediated suppressive signaling to enhance antitumor immune activity [15,16,17,18]. In 2016, the US Food and Drug Administration (FDA) approved two PD-1 monoclonal antibodies, nivolumab and pembrolizumab, for the treatment of platinum-resistant recurrent metastatic HNSCC. The affirmative results from the CheckMate 141 and KEYNOTE-048 trials validated that the PD-1 inhibitors improve survival and response in HNSCC patients compared to the single agent or single agent with chemotherapy group [19,20]. Due to the disruption of T-cell signaling, immune tolerance induction, and immune evasion, HNSCCs are considered to be extremely immunosuppressed malignancies [15,21]. With an overall response rate (ORR) of 15–20%, a limited number of advanced HNSCC patients benefit from immune checkpoint inhibitors. Although a large group of patients do not take advantage of the clinical advantages of ICI therapy, it is valuable to identify valid predictive biomarkers and advanced treatment approaches for most patients [22]. There are more attractive targets for immunotherapy in HNSCC, including the infiltration of immunosuppressive cells, such as regulatory T cells (Tregs), and the upregulation of co-inhibitory receptors on T-cells, such as PD-1 and cytotoxic T lymphocyte-associated protein 4 (CTLA-4) [23,24]. Furthermore, the tumor microenvironment (TME) of the HPV-associated subset of HNSCC has a distinct immune cell profile that is a suitable antigenic target [25]. Below, we summarize the immune responses in HNSCCs and explore various immunotherapy strategies against them, including oncolytic immunotherapy, monoclonal antibodies, and vaccines. We also further address some of the challenges in each strategy.

## 2. Role of the Immune Response in the Development and Progression of HNSCC

The complex TME of HNSCC encompasses promiscuous cellular and molecular components that are interrelated with prognosis in HNSCC patients. Among those entities, the immune system plays an indispensable role during tumor initiation, development, and progression. Immunosuppression has been reported to be one of the main risk factors (0.2% to 1%) for the development of HNSCC and the prognosis is relatively poor [26]. The failure of the immune response to cancer cells may be attributed to the formation of inhibitory immune cells and the secretion of repressive cytokines and mediators, ultimately leading to immune escape [27].

The inhibitory immune cells, such as Tregs, tumor-associated macrophages, and myeloid-derived suppressor cells (MDSCs) are demonstrated to enhance cancer progression and immune escape in HNSCC [28,29]. Tregs are a subtype of CD4^+^ T cells that prevent autoimmune diseases [30]. There are two groups of Tregs: thymus-derived Tregs (tTregs) and periphery-derived Tregs (pTregs). The Forkhead Box P3 (FOXP3) is a key transcription factor and accountable for the function of Tregs [31,32]. Tregs are assembled into the TME in response to chemokines released by tumor cells and macrophages, which in turn act to suppress the antitumor immune responses [30]. Tregs directly inactivate TAA-specific effector T cells by releasing a variety of immunosuppressive cytokines (IL-10, IL-35, and TGF-β), attenuating the secretion of IL-2, which is essential for effector T cells’ survival, and ultimately inhibiting the immune responses against tumors. Alternatively, Tregs also express CTLA-4 which diminishes co-stimulatory signals to effector T cells by downregulating CD80/CD86 in dendritic cells [33]. Tregs can also suppress the function of various other immune cells, such as NK, NKT, and B-cells [34]. Based on the mechanisms mentioned above, it is not surprising that a high infiltration rate of Tregs in tumor sites is highly indicative of fast tumor progression, recurrence, and an elevated metastasis rate [35]. The ratio between different subgroups of T-cells thus provides an informative measure for tumor occurrence and progression, for example, the CD8^+^/CD4^+^ and CD8^+^/FOXP3^+^ ratios are the most-used measurement for the potency of anti-tumor immune activity [36]. Specifically, the above two subset ratios are only considered tumor suppressive if the CD8^+^ T cells are increased compared to FOXP3^+^ and/or CD4^+^ T cells after RT or ICI treatment. Between CD8^+^/CD4^+^ and CD8^+^/FOXP3^+^ ratios, the latter is more efficient in terms of antitumor immune activity. Considering that one of the most important functions of Tregs is to suppress other immune cells, the high CD8^+^/FOXP3^+^ ratio denotes an escape of tumor cells from the immune surveillance [37]. Jie et al. also found that intra-tumoral FOXP3^+^ Tregs in HNSCC patients produce an immunosuppressive TME through the upregulation of immune checkpoint receptors [38]. The high-level infiltration of Tregs was remarkably associated with shorter overall survival (OS) in most solid tumors, including melanoma, renal, cervical, and breast cancers; however, opposite results were obtained in colorectal, esophageal, and head and neck tumors [39]. Recent reports demonstrated that FOXP3^+^ tumor-infiltrating lymphocytes (TILs) were associated with improved overall survival (OS) in HNSCC patients (HR: 0.80; 95% CI: 0.70–0.92) [40,41]. Another study showed that both a high proportion of CD4+ (HR: 0.77; 95% CI: 0.65–0.93) and a high proportion of CD8+ TILs (HR: 0.64; 95% CI: 0.47–0.88) significantly reduced the risk of death and improved overall survival in HNSCC patients [42].

Myeloid-derived suppressor cells (MDSCs) are a group of diverse cells that arise from the bone marrow (BM) and undergo numerous stages of differentiation until they evolve into macrophages, neutrophils, and dendritic cells (DCs) [43]. However, MDSCs are unnaturally generated, activated, and differentiated under pathological conditions, such as cancer [44]. MDSCs notably restrict the antitumor activity of T-cells, particularly cytotoxic T lymphocytes (CTLs). Moreover, MDSCs reduce the action of proinflammatory cells such as natural killer (NK) cells, DCs, and B-cells. On the other hand, MDSCs enhance the production of anti-inflammatory Tregs, Tumor-associated macrophages (TAMs), as well as Th17 cells, which reshape the tumor-promoting TME [45]. The tumor-promoting MDSCs can also prepare a microenvironment or a premetastatic niche (pMN), enhance angiogenesis by secreting matrix metalloprotease 9 (MMP9), and promote the tumor mesenchymal-epithelial transition (MET) to facilitate tumor growth [46,47,48]. Recently, Pang et al. reported that the number of MDSCs was increased in the tissues of patients with oral squamous cell carcinoma (OSCC), and this elevated expression was positively associated with lymph node metastasis, the pathological grade, the T stage, and poor prognosis [49].

Tumor-associated macrophages (TAMs) are macrophages that infiltrate the tumor microenvironment (TME) (Figure 1). TAMs play a crucial role in oral carcinogenesis by influencing tumor growth, progression, and metastasis through immunosuppression. There are two types of TAMs: the classically activated M1 phenotype with anti-tumor activity, and the alternatively activated M2 phenotype, that possesses tumor-promoting activity [50]. M1-like TAMs are stimulated by tumor necrosis factor α (TNF-α), interferon-γ (IFN-γ), and granulocyte–macrophage colony-stimulating factor (GM-CSF), expressing CD40, CD80, and CD86 surface markers. On the other hand, M2-like TAMs expressing CD163, CD204, and CD206 are triggered by transforming growth factor beta (TGF-β) and IL-10 [51]. Interestingly, CD86 is found to be expressed on both types of macrophages [52]. A meta-analysis revealed that TAMs and M2-like macrophages exhibited an increased density in the TME of HNSCC patients, and were associated with vascular and lymphatic invasion, nodal involvement, and an advanced T stage [53]. M2 subtypes can reportedly induce radio-resistance in HPV-HNSCC patients via the activation of EGFR [54]. A study showed that T-cell apoptosis and immunosuppression in OSCC patients were induced by CD163^+^ and CD204^+^ TAM subtypes via IL-10 and PD-L1 [55]. Recently, EGF production by CD206^+^ TAMs was observed to promote proliferation and invasion in OSCC [56]. M2 macrophage-secreted cytokines such as TGF-β, IL-13, and IL-1 reportedly promote tumorigenesis in OSCC [57]. TAMs are also associated with cancer stem cells and poor prognosis in OSCC patients [58].

## 3. Overview of Immune Checkpoint Proteins

Immune checkpoints are important in supporting the proper function of the immune system. Several immune checkpoint signaling pathways are especially essential in the tumor immune microenvironment, consisting of programmed death receptors and their ligands such as PD-L1 and PD-L2, as well as those which are stimulatory (e.g., CD40L and CD70) and inhibitory (e.g., PD-1, CTLA-4, LAG-3, and TIM-3) [59]. Effector T cells lose their ability to eliminate tumor cells when a programmed death receptor engages with its ligand, resulting in an escape from host immune surveillance. In this regard, immune checkpoint inhibitor (ICI) therapy aims to obstruct the interaction between T-cell surface-expressed PD-1 and its ligand, PD-L1, expressed by the tumor cell to enhance the host’s immunity against tumor cells [60,61]. ICIs specifically target a single pathway while others might remain active, thus, it is important to consider combining other target therapies to achieve synergistic anti-tumor effects, such as multiple ICI therapies. CTLA-4 is also a typical immune checkpoint protein; it assists in the activation of the phosphoinositide 3-kinase (PI3K), which in turn affects the CD3ζ (zeta) chain, limiting the signaling potential of the T-cell receptors (TCRs). Other important ICIs include T-cell immunoglobulin mucin-3 (TIM-3), lymphocyte activation gene 3 (LAG-3), and V domain Ig suppressor of T-cell activation (VISTA), and TIGIT (T-cell immunoglobulin and ITIM domain) and Siglec-15 have shown their applications in advanced malignancy such as melanoma, NSCLC, and HNSCC [62,63,64].

### 3.1. Programmed Cell Death Protein-1/Programmed Death Ligand-1 (PD-1/PD-L1)

PD-1 belongs to the CD28 receptor family. It is found primarily on activated or effector T-cells and B-cells and is present in monocytes and a small proportion of thymocytes. Two PD-1 ligands, PD-L1 and PD-L2, are expressed on endothelial and epithelial antigen-presenting cells and activated lymphocytes [65]. PD-L1 is stimulated by an array of inflammatory cytokines, including type I and type II interferons (IFNs), vascular endothelial growth factor (VEGF), and tumor necrosis factor-alpha (TNF-α), whereas PD-L2 is expressed simply on activated macrophages and DCs. The overexpression of PD-L1 on tumor cells may contribute to tumor development [66]. The interaction between PD-1 and PD-L1 is demonstrated to attenuate T-cell receptor-mediated lymphocyte proliferation and cytokine secretion such as IFNs, TNF-α, and VEGF [67]. Moreover, Tregs facilitate PD-1 to engage with PD-L1 and suppress CD4^+^ and CD8^+^ T effector cells in the TME [68]. The PD-1/PD-L1 pathway may become active in a chronic inflammatory environment. Activation through the PD-1/PD-L1 axis plays a particularly important role in the development of HPV + HNSCC; such tissues have increased lymphocytes and higher PD-L1 levels compared to HPV-HNSCC tissues [25]. Depending on the immune milieu of a given HNSCC tumor tissue, the PD-1/PD-L1 axis may be targeted at different levels: The inhibition of PD-1 may prevent binding to PD-L1/PD-L2, while the targeting of PD-L1 may prevent its interaction with PD-1. The interactions between PD-L1 and PD-1 directly regulate the TME and have different functional effects on the actions of T-cells, DCs, BM-derived suppressor cells (MDSCs), and Tregs [69]. The PD-L1 pathway attenuates antitumor T-cell function and affects cellular interactions between the innate and adaptive immune responses, such as between DCs, MDSCs, and Tregs [70].

### 3.2. Cytotoxic T Lymphocyte Antigen 4 (CTLA-4)

The immune checkpoint receptor CTLA-4 is mainly expressed in T-cells, with lower levels of expression seen in active B-cells, monocytes, granulocytes, DCs, and Tregs [71]. In the latter context, the CD28-mediated activation of CTLA-4 triggers the production of the immunosuppressive molecule, transforming growth factor beta (TGF-β) [72]. CTLA-4 and CD28 both serve as transmembrane receptors. CTLA-4 can interact with the B7 protein, leading to T-cell dysfunction and contributing to the downregulation of the immune response. Under normal conditions, the immunosuppressive function of CTLA-4 helps to balance effective immune responses without unduly damaging healthy tissues. However, cancer cells release transforming growth factor beta (TGF-β), which can increase the expression of CTLA-4, leading to T-cell exhaustion [73]; in the exhausted state, T-cells exhibit decreased functionality and can potentially exert immunosuppressive effects. The binding affinity of CD28 to CTLA-4 on the surface of T-cells exceeds its affinity for co-stimulatory molecules such as CD80 and CD86. Such binding inhibits the proliferation and functionals of T-cells [74].

### 3.3. T-Cell Immunoglobulin Mucin-3 (TIM-3)

TIM-3 was first described in 2002 as being expressed on CD4^+^ Th1 lymphocytes but not Th2 lymphocytes [75]. It is also found in Tregs, DCs, monocytes, mast cells, NK cells, and tumor-infiltrating lymphocytes (TILs) [76]. Research has shown that TIM-3 and its ligands play a role in regulating T-cell tolerance. A major TIM-3 ligand is galectin-9 [77]. Upon binding to this member of the galectin family, TIM-3 can inhibit the proliferation of Th1 and Th17 cells, leading to the apoptosis of Th1 cells, the reduction of CD8^+^ T cell function, the significant proliferation of MDSCs, and the muting of the immune response [78]. A monoclonal antibody-induced blockade of TIM-3 was reported to enhance the IFN-γ-mediated anti-tumor response of T-cells [79]. Increased TIM-3 expression contributes to effector T-cell exhaustion, which may lead to an ineffective antitumor immune response and hinder tumor eradication. This scenario may contribute to cancer metastasis and recurrence in patients with HNSCC [80].

### 3.4. Lymphocyte Activation Gene 3 (LAG-3)

The surface molecule LAG-3/CD223, which belongs to the immunoglobulin superfamily, was first discovered in 1990 [81]. It is mainly observed in activated T-cells but can also be found in NK cells, B-cells, and plasmacytoid DCs. LAG-3 can negatively modulate T-cell proliferation, activation, and homeostasis [82], and thus shares functions similar to those of CTLA-4 and PD-1 [83]. LAG-3 critically contributes to the suppressive function of Tregs [84]. In certain malignancies, the simultaneous expression of LAG-3 and PD-1 on TILs is associated with impaired CD8^+^ effector T-cell function, contributing to tumor immune evasion [85]. LAG-3 is typically found on Tregs and is essential for maintaining the T cell balance promoted by Tregs. The blockade of LAG-3 can impede Treg activation and abrogate Treg-mediated suppression. Interestingly, non-Treg CD4^+^ T cells that also express LAG-3 contribute to the inhibitory capabilities. LAG-3 expression is increased on CD4^+^-FOXP3^+^ and IL-10-secreting type 1 (Tr1) regulatory T cells. In HNSCC, LAG-3 is highly expressed on Tregs and interrelated with poor progression-free survival (PFS) and overall survival (OS) [38,86]. Deng et. al. also showed that the overexpression of LAG-3 in human primary HNSCC correlates with elevated pathological categories, massive tumor sizes, and pragmatic lymph node levels. LAG-3 is usually overexpressed on TILs and OS analysis pinpoints that LAG-3 acts as a predictive element unaided by tumor size [87]. The significance of LAG-3-targeting has been illustrated in 108 clinical trials, including anti-LAG-3 monoclonal antibodies, bispecific molecules, fusion proteins, and CAR-T cells [88]. It is also found that in HNSCC, LAG-3 expression could increase in the extracellular vesicles by miRNAs such as miR-7704 and miR-21-5p, suggesting that LAG-3 is a promising target for cancer [89].

### 3.5. Glucocorticoid-Induced TNFR Family-Related Protein (GITR)

GITR is present on the surfaces of CD25^+^-CD4^+^ Tregs, effector T cells, and NK cells [90]. The binding of GITR to its ligand, GITRL, can decrease Treg recruitment, abrogate their inhibitory effect, and trigger MAPK/ERK and NF-κB signaling, thereby increasing the proliferation and differentiation of T-cells, promoting the secretion of pro-inflammatory cytokines, and enhancing antitumor function [91,92]. Thus, GITR acts as an immune checkpoint protein. This has been leveraged by immunotherapy with an anti-GITR antibody (DTA-1 clone), which prevents the suppression of Tregs [93].

### 3.6. V-Domain Ig Suppressor of T-Cell Activation (VISTA)

VISTA is a recently identified checkpoint molecule that shares functional similarities with PD-L1 and can effectively suppress T-cell activation. In mice, VISTA is highly expressed on TILs and its blockade enhances antitumor immunity in various tumor models [94]. VISTA is mainly expressed on myeloid APCs and T-cells, especially Tregs [95]. VISTA contributes to creating an immunosuppressed TME by promoting Treg maturation and inhibiting T-cell activation. Both VISTA and PD-L1 exert their inhibitory effects on T-cell activation through various immunoregulatory networks and contribute to modulating T-cell responses [96]. VSIG3, which is a novel ligand for VISTA, mediates homogeneous adhesion in a calcium-independent manner [97,98]. The simultaneous interaction of VSIG3 and VISTA on activated T-cells inhibits the proliferation of T-cells and promotes the synthesis of cytokines and chemokines. The inhibitory effect of VSIG3 on activated T- cells and its striking expression in colorectal, hepatocellular, and gastric carcinomas suggests that targeting the VSIG3/VISTA pathway is a promising innovative approach for cancer immunotherapy [97].

## 4. Immune Checkpoint Inhibitor-Based Therapeutics in HNSCC

### 4.1. Targeting PD-1/PD-L1

The use of PD-1/PD-L1 targeting monoclonal antibodies to counteract the immunosuppressive role of PD-1 could help maintain a robust T-cell response against tumor cells [99] (Figure 2). The USA FDA has approved five monoclonal antibodies targeting PD-1 and PD-L1. Two of them target PD-L1, Atezolizumab, and Durvalumab, while Nivolumab, Pembrolizumab, and Cemiplimab are against PD-1. Nivolumab and Pembrolizumab are approved for patients with relapsed or metastatic cisplatin-resistant HNSCC [100], whereas Cemiplimab exhibits significant responses in patients with metastatic cutaneous squamous cell carcinoma [101].

Nivolumab is an IgG4 monoclonal antibody that acts as an anti-PD-1 agent and blocks co-suppressive signals via the PD-1/PD-L1 signaling pathway. It was a significant milestone as the first FDA-approved immunotherapy for patients with HNSCC, based on the results of a Phase III clinical trial known as CheckMate 141 (NCT02105636) [102]. The study involved 361 participants with recurrent HNSCC who had experienced progression 6 months after platinum chemotherapy. These individuals were divided into two groups in a 2:1 ratio, with one group receiving nivolumab and the other receiving standard systemic drug therapy with methotrexate, docetaxel, or cetuximab every two weeks. The primary endpoint of the study was overall survival (OS), while secondary objectives were objective response, progression-free survival (PFS), safety, and patient-reported quality of life [19]. The study results showed that the group treated with nivolumab survived for a median of 7.5 months, while the standard treatment group achieved survival of 5.1 months. The estimated one-year survival rates for the nivolumab and standard treatment groups were approximately 36.0% and 16.6%, respectively. At 6 months, the PFS rate was 19.7% in the nivolumab group, compared with 9.9% in the standard treatment group. Furthermore, the incidence of severe adverse effects (grade III or IV) was lower in the nivolumab group (13.1%) compared to the standard treatment group (35.1%) [102].

Pembrolizumab is a humanized IgG4-κ monoclonal antibody with a high affinity for PD-1. It received its first FDA approval in 2017 following results from the phase Ib cohort expansion study, KEYNOTE 012 [103]. Importantly, in advanced HNSCC patients, treatments with pembrolizumab have shown high survival rates (6-month OS rate: 58%; and 12-month OS rate: 38%) and no adverse event-related deaths, an outcome rarely seen with existing cytotoxic or targeted therapies [104].

The comparison between pembrolizumab and standard treatment has been made in KEYNOTE 040 to validate that it prolongs the median OS to 8.4 months from 6.9 months for HNSCC patients [105]. In addition, a phase III clinical trial called KEYNOTE 048 evaluated the use of pembrolizumab for the treatment of relapsed or metastatic HNSCC and reported significantly improved treatment outcomes with pembrolizumab in 2019. In an interim analysis, the combination of pembrolizumab and chemotherapy was found to result in longer OS than the combination of cetuximab and chemotherapy (13.0 months vs. 10.7 months) in the general patient population. Based on these compelling efficacy and safety results, pembrolizumab plus chemotherapy has become the primary treatment option for patients with recurrent or metastatic HNSCC. However, pembrolizumab monotherapy has become the first-line treatment for patients with recurrent or metastatic PD-L1^+^ HNSCC [20].

Emerging data have suggested that PD-L2 is an alternative ligand for PD-1, especially in PD-L1-negative patients. The interaction between PD-1 and PD-L2 exhibits a strong suppression of T cell receptor (TCR)-expressed proliferation and a reduction of cytokine production such as IL-4, IL-10, and INF-γ by CD4^+^ T cells. Moreover, the expression and function of PD-L2 appear to be comparable to PD-L1 and both can reduce PD-1 signaling affecting T-cell proliferation [106]. In addition, Yearley et al. used a unique immunohistochemistry assay to investigate the expression of PD-L2 in tumor tissues and the correlation between clinical response and PD-L2 status in tumor tissues from patients with relapsed/metastatic (R/M) HNSCC treated with pembrolizumab. The results indicated that there was a highly significant association between PD-L2 and PD-L1 in these tumors (*p* < 0.0001) [107]. However, PD-L2 expression was also detected in tumors without PD-L1 expression, suggesting that the predictive potential of PD-L2 is not dependent on PD-L1 [108].

Atezolizumab is an anti-PD-L1 monoclonal antibody that blocks PD-L1 and generates an anti-cancer immune response by blocking PD-L1 [109]. In 2014, it was used against HNSCC and other tumor types such as NSCLC, melanoma, gastric cancer, colorectal cancer, and pancreatic cancer to investigate safety and tolerability. This first Phase I study (PCD4989g; NCT01375842) did not follow the traditional clinical trial design [110]. Later, in 2018, A. D. Colevas et. al. reported on the safety and long-term efficacy of atezolizumab as a single agent in a Phase Ia clinical trial in patients with HNSCC. This Phase Ia study involved 32 patients in whom treatment-related adverse events (TRAEs) of varying grades, including grades 1, 2, 3, and 4, were observed. There were no deaths and no grade 5 TRAEs. Overall survival was 6.0 months with good tolerability of atezolizumab [111].

Cemiplimab is an IgG4 monoclonal antibody with high affinity and dynamics against PD-1. A study in 2018 has shown that cemiplimab is effective in reducing tumors in the phase I advanced cutaneous squamous cell carcinoma (CSCC) expansion cohort (NCT02383212) and the metastatic cohort of the phase II trial (NCT02760498) [112]. Furthermore, cemiplimab is reported to exhibit marked anti-tumor activity with a fixed dose (350 mg intravenously every 3 weeks), and durable long-term effects (3 mg/kg intravenously every 2 weeks) in metastatic CSCC (NCT02760498) [113].

Durvalumab is a monoclonal antibody that directly engages with PD-L1 with high affinity and, as a result, blocks the interaction with PD-1 and CD80. In the HAWK Phase II study, it was administered to individuals with recurrent and/or metastatic HNSCC who had PD-L1 expression in more than 25% of tumor cells after unsuccessful platinum-based chemotherapy [114].

### 4.2. Targeting CTLA-4

Ipilimumab is a monoclonal antibody (mAb) targeting CTLA-4 (Figure 2) that has been approved by the FDA for the treatment of metastatic melanoma. However, there are significant concerns about potential serious toxicity, including life-threatening colitis. Ongoing clinical trials are evaluating the use of ipilimumab in combination with cetuximab and intensity-modulated radiotherapy (IMRT) in patients with advanced HNSCC (NCT01860430 and NCT01935921). Concurrently, a phase I study is ongoing to evaluate the safety and optimal dosing of MGA271 (also known as enoblituzumab), a humanized mAb targeting CD276 (B7-H3), in combination with ipilimumab. This study includes patients with B7-H3-expressing solid tumors, including HNSCC (NCT02381314). Another anti-CTLA4 antibody, tremelimumab, is also being studied in clinical trials in combination with the anti-PD-L1 antibody, durvalumab [99]. A randomized phase II clinical trial with 267 R/M HNSCC patients showed that the combination of durvalumab and tremelimumab resulted in clinically relevant overall survival and manageable toxic effects [115]. The clinical trial CheckMate 651 (NCT02823574) comprehensively demonstrated that a combination of nivolumab and ipilimumab is an excellent disease control agent in R/M HNSCC, increasing the median OS from 13.5 to 13.9 months compared to the EXTREME therapy [116].

### 4.3. Targeting GITR

AMG 228 is a human IgG1 monoclonal antibody that acts as an agonist by binding to human GITR, a molecule expressed by regulatory T cells. In a phase I study (NCT02437916), 30 patients were selected with advanced solid tumors, including colorectal cancer, HNSCC, urothelial transitional cell carcinoma, non-small cell lung cancer (NSCLC), and melanoma. They administered AMG 228 intravenously every 3 weeks. By employing a two-stage dose escalation strategy: first, a single patient cohort received 3, 9, 30, or 90 mg, followed by a “rolling 6” design involving two to six patients till the maximum tolerated dose or an elevated planned dose of 1200 mg was reached. The primitive focus of the study was to estimate the safety, pharmacokinetics, pharmacodynamics, and maximum tolerated dose (MTD) based on the patients’ responses. The outcomes of the study concluded that within this patient population, AMG 228 was well tolerated up to 1200 mg. No dose-limiting toxicity (DLT) was observed, and the highest administered dose did not reach the maximum tolerated dose (MTD). However, there was no observable evidence of T-cell activation or any antitumor activity following a single administration of AMG 228 therapy [117].

## 5. Combined Immune Checkpoint Inhibitor Therapy in Oral Cancer

### 5.1. PD-1/CTLA-4 Combination

The immune checkpoint inhibitors that have been used alone/combined or in combination with other therapies are listed in Table 1. In a documented clinical case report, an unprecedented approach involving the combination of the anti-PD-1 antibody nivolumab and the anti-CTLA4 antibody ipilimumab was administered to a surgical patient diagnosed with HNSCC. Following, a 3-week course of treatment with these combined drugs, a computed tomography examination revealed a positive response in the patient’s cancer condition. However, subsequent magnetic resonance imaging (MRI) showed local recurrence at 7 months after the combined treatment. Notably, the expression levels of PD-L1 and CTLA-4 exhibited significant decreases at the cervical lymph node, whereas no substantial alteration was observed in the expression level of PD-L2 or the quantities of various immune cells, including B-cells, T-cells, T helper (Th) cells, cytotoxic and regulatory T lymphocytes, and NK cells [118]. This case report suggests that combined PD-1/CTLA-4 immune checkpoint inhibitor therapy may be effective in some patients with HNSCC, but further research is needed to confirm its efficacy and safety. More research is also needed to understand why the patient’s cancer recurred despite decreases in the expression levels of PD-L1 and CTLA-4.

### 5.2. PD-1/GITR Combination

Clinical studies have shown that the combination of anti-GITR and anti-PD-1 antibody therapies can enhance the antitumor activity of T-cells. Among the anti-GITR antibodies, BMS-986156 is an agonistic human IgG1 monoclonal antibody that activates effector T cells and may also reduce or inactivate Tregs [119]. A phase I/IIa clinical trial (NCT02598960) evaluated the safety and efficacy of BMS-986156 alone and in combination with the anti-PD-1 antibody, nivolumab, in patients with advanced solid tumors. The trial involved a dose-escalation design, with 66 patients receiving BMS-986156 (10–800 mg) or BMS-986156 (30–800 mg) plus nivolumab (240 mg) every 2 weeks. The study results showed that BMS-986156 ± nivolumab was well tolerated by patients and no dose-limiting toxicity occurred. Significant antitumor activity was seen when BMS-986156 was combined with nivolumab at doses predicted to have biological activity [120].

## 6. Immune Checkpoint Inhibitors in Combination with Other Therapies

### 6.1. Radiotherapy

Radiotherapy (RT) is widely used to treat solid tumors, with more than 50% of patients receiving this modality. RT has the advantage over chemotherapy in minimizing systemic toxicity [121]. When RT is combined with immune checkpoint inhibitors (ICIs), it can potentiate the synergistic effects, where RT contributes to the normalization of the tumor vascular system, enhance the expression of leukocyte adhesion molecules on endothelial cells, and stimulate the secretion of chemokines that attract CD8^+^ T cells [122]. ICI treatment is intended for recurrent or metastatic HNSCC that has previously responded to RT, especially for those whose immune cells are being transformed into immunosuppressive and radio-resistant phenotypes. In a phase II study (NCT02641093), the employment of pembrolizumab as adjuvant RT has been shown to improve the survival of patients with locally advanced HNSCC [123]. In some patients, following RT can prompt somatic mutations, which can lead to the development of new tumor-associated antigens (TAAs) that can be targeted for more robust immune responses. The concurrent administration of RT and ICIs such as nivolumab (NCT02684253 and NCT03349710) is generally safe with no significant immune-related adverse events on HNSCC [124].

### 6.2. Chemotherapy

Tumor-related antibodies are currently used primarily in combination with chemotherapy due to their limited efficacy as monotherapies [125]. Chemotherapy can potentiate the efficacy of immune checkpoint inhibitor (ICI) therapy by promoting the release of neoantigens, modifying the TME through the depletion of Tregs and MDSCs, and reducing PD-L2 expression on DCs and tumor cells. Additionally, chemotherapy (CT) can stimulate APC maturation and increase MHC-I expression [126]. One example of an approved combination regimen is the use of pemetrexed and carboplatin chemotherapy with pembrolizumab, where pembrolizumab exhibits minimal overlapping toxicity with the chemotherapeutic agents [127]. Other combination regimens under investigation include the use of ipilimumab, paclitaxel, and carboplatin in stage-IV non-small-cell lung carcinoma (NCT02279732) [67] and nivolumab with ipilimumab as the standard of care for the first-line treatment of HNSCC (NCT02741570) [128]. However, it is important to note that chemotherapy can induce side effects that may interfere with the action mechanisms of ICIs, such as by inhibiting the clonal expansion of effector lymphocytes. Therefore, the potential impact of chemotherapy must be carefully assessed when designing treatment strategies.

## 7. Discussion and Future Perspectives on Oral Cancer Immune Therapy

Traditional radiotherapy and chemotherapy are still the most effective treatment for later-stage oral cancer. After surgery, radiotherapy is usually the standard treatment, while chemotherapy may be used to prevent recurrence. However, these traditional treatments can cause complications, such as difficulty swallowing, neck stiffness, blood disorders, and so on, that can interfere with work and family life. Immunotherapy has emerged as one of the most advanced methods in cancer treatment today. Although remarkable progress has been made in the immunotherapy of HNSCC, a newly identified pattern of cancer progression has been discovered, known as hyperprogressive disease (HPD), in which patients rapidly deteriorate in the early stages of treatment. Therefore, further studies are needed to comprehensively explore other possible therapeutic strategies in HNSCC [129]. It is, therefore, worth investigating this natural immunological phenomenon to develop better therapeutic strategies against oral cancer. To improve and optimize the efficacy and safety of future HNSCC therapies, several advanced strategies are currently being investigated, for example, combination therapy, dendritic cell vaccines, targeted therapy, peptide therapy, and precision immunotherapy. Two major approaches of precision medicine such as multi-OMICS and a personalized preclinical platform are particularly considered for the individualized treatment of HNSCC patients with improved efficacy. The first approach can assist in the identification of tumor behavior, employing biomarkers. Only a very small group of patients with R/M HNSCC can benefit from ICIs due to a lack of validated biomarkers. The selection of appropriate biomarkers may facilitate the achievement of clinically meaningful therapeutic feedback. The second approach could predict drug sensitivity in clinical samples. Preclinical cancer models such as patient-derived xenografts (PDX) have remarkable parallels at the genetic, transcriptomic, and proteomic levels that may strengthen biomarker-based clinical trials and improve the management of HNSCC [130,131].

Personalized immunotherapy can provide better clinical outcomes compared to conventional therapies. It can control the individual characteristics of each tumor [132]. Current proteomics techniques can be employed to assess immune profiling in HNSCC patients, providing insights into new biomarkers and mutation sites. The information of individual immune profiling may also provide a personalized treatment plan to achieve effective health outcomes. Although ICI treatment with a single agent leads to a long-lasting effect in advanced-stage cancer patients, only a small group of patients benefit from it, while combination immunotherapy maximizes the benefit for the majority of patients by reducing resistance to single agents [133,134]. Another perspective to overcome the resistance mechanisms of single-agent immunotherapy is to combine it with targeted therapy. Targeted therapy can minimize damage to healthy cells while specifically targeting and killing cancer cells. So, when ICI is combined with targeted therapy, it may impede critical molecular pathways that help cancer cells grow and invade, besides engaging with the immune system. This combined therapy may amplify anti-cancer efficacy while reducing immunosuppressive side effects [135]. In some cases, immune checkpoint inhibitors may induce limited efficacy and adverse effects with CT and/or RT, thus other safe and effective treatment approaches should be considered. Peptides are bioactive molecules with distinct properties, such as lower affinity and a shorter half-life in the body compared to antibodies, thereby enabling them to selectively bind to certain surface proteins of cancer cells, subsequently blocking their biological functions [136]. In addition, peptides in combination with immune checkpoint inhibitors may increase the specificity and efficacy of cancer immunotherapy by targeting specific tumor antigens [137]. Thus, peptides in combination with immune checkpoint inhibitors may enhance the efficacy of cancer immunotherapy by targeting specific tumor antigens. The clinical results from the bioavailable immune checkpoint inhibitors combined with other therapies will presumably stimulate next-generation immunotherapeutic agent development. While remarkable progress has been made in the development of ICIs, a great number of patients still suffer from severe and chronic immune-related adverse effects. Thus, it is imperative to identify predictive biomarkers that will help physicians determine who could be most likely to benefit from immunotherapy. Currently, a few examples of predictive biomarkers in the immunotherapy for HNSCC have been reported. The hypomethylation of the CTLA4 promoter is associated with a positive response to ICIs and prolonged progression-free survival, serving as a predictive biomarker in HNSCC [138]. The presence of infiltrating T cells in the tumor containing interferon-gamma (IFN-γ) responsive genes, a key factor for PD-L1 expression in HNSCC, may enhance the efficacy of anti-PD-1 therapies [139]. Furthermore, CCND1 amplification is mostly associated with an unfavorable response to ICIs and an unfavorable prognosis in HNSCC [140].

## 8. Conclusions

In recent years, attention to cancer immunotherapy for head and neck squamous cell carcinoma (HNSCC) has increased significantly due to a better understanding of cancer immunology. This innovative approach is reshaping the treatment landscape for HNSCC. Compared to conventional treatments, both immune monotherapy and combination therapy are effective in reducing morbidity and prolonging the survival of HNSCC patients. However, there are still many challenges in the field of cancer immunotherapies for HNSCC, and we are still in the early stages of understanding the optimal integration of surgery, chemotherapy, and radiotherapy with immunotherapy. Therefore, challenges such as tumor immune resistance, immune escape, and immune-related adverse events need to be investigated to gain a clear understanding of the basis of immunomodulatory mechanisms in HNSCC and advances the field. In addition, selecting an appropriate therapeutic strategy, identifying predictive biomarkers, understanding the patient’s histology-specific considerations, and predicting the clinical response are critical to improving the therapeutic efficacy of cancer immunotherapy.

## Figures and Tables

**Figure 1 cells-13-00413-f001:**
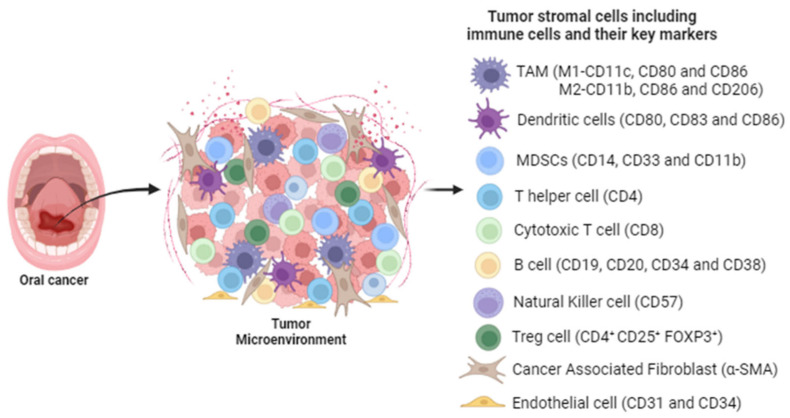
Tumor stromal cells, including immune cells and non-immune cells, and their key markers. Key immune cells in the TME include Macrophages, Dendritic cells, MDSCs, Tcells, and Bcells. The non-immune cells include cancer-associated fibroblasts and endothelial cells. This figure was created with BioRender.com (accessed on 12 October 2023).

**Figure 2 cells-13-00413-f002:**
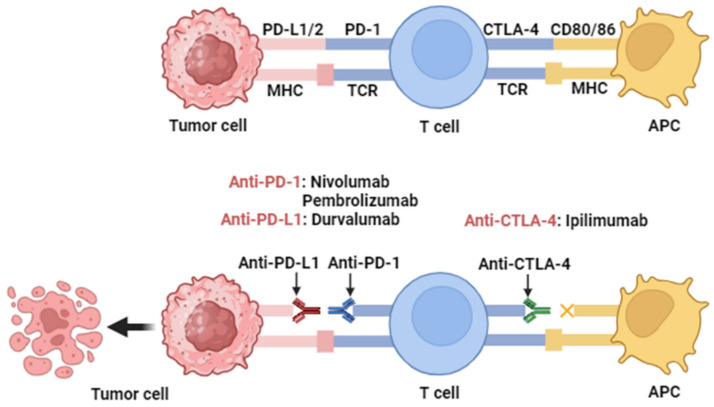
Action mechanisms of antibodies targeting PD-1, PD-L1, and CTLA-4. Immune checkpoint proteins PD-1 and PD-L1 are expressed in T cells, B cells, and antigen-presenting cells (APCs). High levels of PD-L1 expression hide cancer cells from T cells and enable them to grow unchecked. When PD-1 binds to PD-L1, T cells stop dividing and killing cancer cells. Through activation of PD-1/PD-L1 signaling, cancer cells evade immune responses. Inhibition of the PD-1/PD-L1 interaction by monoclonal antibodies overturns this process and augments activity. This allows T cells to become activated and kill cancer cells. This figure was created with BioRender.com (accessed on 10 October 2023).

**Table 1 cells-13-00413-t001:** List of completed ICIs clinical trials for the treatment of head and neck cancers.

Serial	ICIs	Target	Phase	Arm/Group	No. of Patients	Combination	Clinical Trial Number	Primary Endpoint	Study Start/Date
1	Durvalumab	PD-L1	I	Recurrent and/or metastatic head and neck squamous cell carcinoma	71	Tremelimumab	NCT02262741	Safety, tolerability, antitumor activity, PK, pharmacodynamics, and immunogenicity	15/10/2014
2	Pembrolizumab	PD-1	II	Recurrent and/or metastatic head and neck squamous cell carcinoma	172	After platinum-based and cetuximab therapy	NCT02255097	Objective response rate (ORR)	24/10/2014
3	Durvalumab	PD-L1	I/II	Head and neck cancer solid tumors	176	Epacadostat	NCT02318277	Safety, tolerability, pharmacokinetics, immunogenicity, and preliminary efficacy	05/01/2015
4	Pembrolizumab	PD-1	III	First-line treatment of recurrent and/or metastatic head and neck squamous cell carcinoma	882	Chemotherapy	NCT02358031	Progression-free survival (PFS)	19/03/2015
5	Pembrolizumab	PD-1	II	Advanced head and neck squamous cell carcinoma	78	Acalabrutinib	NCT02454179	Overall response rate (ORR)	01/05/2015
6	Nivolumab	PD-1	II	Metastatic head and neck squamous cell carcinoma (HNSCC)	65	Stereotactic body radiotherapy (SBRT)	NCT02684253	Best overall response (BOR)	11/02/2016
7	Pembrolizumab	PD-1	I/II	Locally advanced laryngeal squamous cell carcinoma (Grade 3/4)	9	Chemotherapy	NCT02759575	Adverse effects	01/04/2016
8	Pembrolizumab	PD-1	I	Recurrent and/or metastatic head and neck squamous cell carcinoma	36	Talimogene laherparepvec	NCT02626000	Safety and toxicity	06/04/2016
9	Nivolumab	PD-1	I	Intermediate and high-risk local regionally advanced head and neck cancer	40	Chemotherapy	NCT02764593	Safety	01/06/2016
10	Nivolumab	PD-1	II	Recurrent or metastatic squamous cell carcinoma	425	Ipilimumab	NCT02823574	Overall response rate (ORR)	08/11/2016
11	Nivolumab	PD-1	I/II	Neoadjuvant to surgery in advanced stage head and neck squamous cell carcinoma (HNSCC)	33	Ipilimumab	NCT03003637	Feasibility and safety	28/02/2017
12	Nivolumab	PD-1	II	Recurrent or metastatic salivary gland carcinoma	98	-	NCT03132038	Progression-free survival (PFS): after 6 months of treatment	24/03/2017
13	Nivolumab	PD-1	III	Recurrent and/or metastatic head and neck squamous cell carcinoma	124	-	NCT05802290	Adverse effects	27/11/2017
14	Nivolumab	PD-1	III	Locally advanced squamous cell carcinoma	74	Cisplatin in combination with radiotherapy	NCT03349710	Event-free survival (EFS)	15/12/2017
15	Pembrolizumab	PD-1	II	Recurrent and/or metastatic head and neck squamous cell carcinoma	29	Afatinib	NCT03695510	Toxicities and efficacies	24/01/2019
16	Durvalumab	PD-L1	I/II	Head and neck squamous cell carcinoma	33	-	NCT03829007	Treatment regimen	15/04/2019
17	Pembrolizumab	PD-1	II	First-line treatment of metastatic or unresectable, recurrent head and neck squamous cell carcinoma	18	Ulevostinag	NCT04220866	Safety and efficacy	04/03/2020
18	Ipilimumab	CTLA-4	I	Head and neck cancer (Stage III-IVB)	19	Cetuximab with intensity-modulated radiation therapy	NCT01935921	Side effects and dosage regimen	09/04/2013
19	Ipilimumab	CTLA-4	I	Head and neck squamous cell carcinoma	80	Pembrolizumab	NCT01986426	Safety, tolerability, PK, and efficacy	01/11/2013
20	Ipilimumab	CTLA-4	I/II	Advancedstage head and neck squamous cell carcinoma	33	Nivolumab (neoadjuvant to surgery)	NCT03003637	Feasibility and safety	28/02/2017
21	Ipilimumab	CTLA-4	I	Head and neck cancer (Stage IVA-B)	24	Nivolumab with radiotherapy	NCT03162731	Side effects	11/05/2017

## Data Availability

No new data were created or analyzed in this study. Data sharing does not apply to this article.

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
