# Peer review of "The Use of Immune Regulation in Treating Head and Neck Squamous Cell Carcinoma (HNSCC)"

_cells, 2024, doi:10.3390/cells13050413_

Round 1

Reviewer 1 Report

Comments and Suggestions for Authors

1.     Please, elaborate on recent studies that explore alternative biomarkers demonstrating potential for assessing enhanced clinical immune responses in Head and Neck Squamous Cell Carcinoma (HNSCC):

Noji R., et al. Comprehensive Genomic Profiling Reveals Clinical Associations in Response to Immune Therapy in Head and Neck Cancer. Cancers. 2022;14:3476. 

Hoffmann, F., et al. CTLA4 DNA methylation is associated with CTLA-4 expression and predicts response to immunotherapy in head and neck squamous cell carcinoma. Clin Epigenet 15, 112 (2023).

Ayers M., et al. IFN-γ-related mRNA profile predicts clinical response to PD-1 blockade. J. Clin. Investig. 2017;127:2930–2940.

2.     There is a deficiency in referencing the mechanism of action for PD-L2:

Latchman Y, et al.. PD-L2 is a second ligand for PD-1 and inhibits T cell activationNat Immunol. (2001) 

3.     Please, mention this study:

Cohen E.E.W., et al. Pembrolizumab versus methotrexate, docetaxel, or cetuximab for recurrent or metastatic head-and-neck squamous cell carcinoma (KEYNOTE-040): A randomised, open-label, phase 3 study. Lancet. 2019;393:156–167. 

4.     Please, include and reference in the manuscript, the overall response rates (ORR) of immune checkpoint inhibitors (ICI) among patients diagnosed with head and neck squamous cell carcinoma (HNSCC).

5.     Please mention in the manuscript the safety and efficacy of anti-PD-1/PD-1 therapy in patients with HNSCC and reference:

Mehra R., et al. Efficacy and safety of pembrolizumab in recurrent/metastatic head and neck squamous cell carcinoma: Pooled analyses after long-term follow-up in KEYNOTE-012. Br. J. Cancer. 2018;119:153–159.

6.     Cemiplimab received FDA approval have been approved for of patients with metastatic cutaneous squamous cell carcinoma (CSCC); please, mention it in the manuscript:

Migden MR et al. PD-1 blockade with cemiplimab in advanced cutaneous squamous-cell carcinoma. N Engl J Med. 2018;379(4):341–51.

Rischin D, et al. Phase 2 study of cemiplimab in patients with metastatic cutaneous squamous cell carcinoma: primary analysis of fixed-dosing, long-term outcome of weight-based dosing. J Immunother Cancer. 2020;8(1).

7.     Please include a reference to this source in the manuscript:

Chocarro L. et al. Cutting-Edge: Preclinical and Clinical Development of the First Approved Lag-3 Inhibitor. Cells. 2022;11:2351. 

8.      Please, consider cite BioRender (for the images), if necessary.

9.     Typo in table 2: “Phage”

10.  Ensure all abbreviations used in the manuscript are fully accounted for in the list; several are currently missing.

Author Response

Reviewer 1#

  1. Please, elaborate on recent studies that explore alternative biomarkers demonstrating potential for assessing enhanced clinical immune responses in Head and Neck Squamous Cell Carcinoma (HNSCC):

Noji R., et al. Comprehensive Genomic Profiling Reveals Clinical Associations in Response to Immune Therapy in Head and Neck Cancer. Cancers. 2022;14:3476. 

Hoffmann, F., et al. CTLA4 DNA methylation is associated with CTLA-4 expression and predicts response to immunotherapy in head and neck squamous cell carcinoma. Clin Epigenet 15, 112 (2023).

Ayers M., et al. IFN-γ-related mRNA profile predicts clinical response to PD-1 blockade. J. Clin. Investig. 2017;127:2930–2940.

Author’s response: Thank you very much for your time and valuable suggestion. In response to your suggestions, we have revised our manuscript as following (page 14, line 528-539). “While remarkable progress has been made in the development of ICIs, a great number of patients still suffer from severe and chronic immune-related adverse effects. Thus, it is imperative to identify predictive biomarkers that will help physicians determine who could be most likely to benefit from immunotherapy. Currently, a few examples of predictive biomarkers in the immunotherapy for HNSCC have been reported. Hypomethylation of the CTLA4 promoter is associated with a positive response to ICIs and prolonged progression-free survival, serving as a predictive biomarker in HNSCC [138]. The presence of infiltrating T cells in the tumor containing interferon-gamma (IFN-γ) responsive genes, a key factor for PD-L1 expression in HNSCC, may enhance the efficacy of anti-PD-1 therapies [139]. Furthermore, CCND1 amplification is mostly associated with an unfavorable response to ICIs and an unfavorable prognosis in HNSCC [140].”

  1. There is a deficiency in referencing the mechanism of action for PD-L2:

Latchman Y, et al.. PD-L2 is a second ligand for PD-1 and inhibits T cell activation. Nat Immunol. (2001) 

Author’s response: Thank you very much for your time and valuable suggestion. In response to your suggestions, we have added the reference and revised our manuscript (page 7, line 326-331). “Emerging data have suggested that PD-L2 is an alternative ligand for PD-1, especially in PD-L1-negative patients. The interaction between PD-1 and PD-L2 exhibits a strong suppression of T cell receptor (TCR)-expressed proliferation and reduction of cytokine production such as IL-4, IL-10, and INF-γ by CD4+ T cells. Moreover, the expression and function of PD-L2 appear to be comparable to PD-L1 and both can reduce PD-1 signaling affecting T cell proliferation [106].”

  1. Please, mention this study:

Cohen E.E.W., et al. Pembrolizumab versus methotrexate, docetaxel, or cetuximab for recurrent or metastatic head-and-neck squamous cell carcinoma (KEYNOTE-040): A randomised, open-label, phase 3 study. Lancet. 2019;393:156–167. 

Author’s response: Thank you very much for your time and valuable suggestion. In response to your suggestions, we have added the reference and revised our manuscript (page 7, line 314-316). “The comparison between Pembrolizumab and standard treatment has been made in KEYNOTE 040 to validate that it prolongs the median OS to 8.4 months from 6.9 months of the HNSCC patients [105].”

  1. Please, include and reference in the manuscript, the overall response rates (ORR) of immune checkpoint inhibitors (ICI) among patients diagnosed with head and neck squamous cell carcinoma (HNSCC).

Author’s response: Thank you very much for your time and valuable suggestion. In response to your suggestions, we have added the reference and revised our manuscript (page 2, line 72 -76). “With an overall response rate (ORR) of 15-20%, a limited number of advanced HNSCC patients benefit from immune checkpoint inhibitors. Although a large group of patients do not take advantage of the clinical advantages of ICI therapy, it is valuable to identify valid predictive biomarkers and advanced treatment approaches for most patients [22].”

  1. Please mention in the manuscript the safety and efficacy of anti-PD-1/PD-1 therapy in patients with HNSCC and reference:

Mehra R., et al. Efficacy and safety of pembrolizumab in recurrent/metastatic head and neck squamous cell carcinoma: Pooled analyses after long-term follow-up in KEYNOTE-012. Br. J. Cancer. 2018;119:153–159.

Author’s response: Thank you very much for your time and valuable suggestion. In response to your suggestions, we have added the reference and revised our manuscript (page 7, line 310 -313). “ Importantly, in advanced HNSCC patients, treatments with Pembrolizumab have shown high survival rates (6-month OS rate: 58% and 12-month OS rate: 38%) and no adverse event-related deaths, an outcome rarely seen with existing cytotoxic or targeted therapies [104].”

  1. Cemiplimab received FDA approval have been approved for of patients with metastatic cutaneous squamous cell carcinoma (CSCC); please, mention it in the manuscript:

Migden MR et al. PD-1 blockade with cemiplimab in advanced cutaneous squamous-cell carcinoma. N Engl J Med. 2018;379(4):341–51.

Rischin D, et al. Phase 2 study of cemiplimab in patients with metastatic cutaneous squamous cell carcinoma: primary analysis of fixed-dosing, long-term outcome of weight-based dosing. J Immunother Cancer. 2020;8(1).

Author’s response: Thank you very much for your time and valuable suggestion. In response to your suggestions, we have added the references and revised our manuscript (page 7, line 349 -355). “Cemiplimab, an IgG4 monoclonal antibody with high affinity and dynamics against PD-1. A study in 2018 has shown that Cemiplimab is effective in reducing tumors in phase I advanced cutaneous squamous cell carcinoma (CSCC) expansion cohort (NCT02383212) and the metastatic cohort of the phase II trial (NCT02760498) [112]. Furthermore, cemiplimab is reported to exhibit marked anti-tumor activity with a fixed-dose (350 mg intravenously every 3 weeks), and durable long-term effects (3 mg/kg intravenously every 2 weeks) in metastatic CSCC (NCT02760498) [113].”

  1. Please include a reference to this source in the manuscript:

Chocarro L. et al. Cutting-Edge: Preclinical and Clinical Development of the First Approved Lag-3 Inhibitor. Cells. 2022;11:2351. 

Author’s response: Thank you very much for your time and valuable suggestion. In response to your suggestions, we have added the reference and revised our manuscript (page 5-6, line 252 -254). “ The significance of LAG-3-targeting has been illustrated in 108 clinical trials, including anti-LAG-3 monoclonal antibodies, bispecific molecules, fusion proteins, and CAR-T cells [88].”

  1. Please, consider cite BioRender (for the images), if necessary.

Author’s response: Thank you and the citation has been added in the figure legends (page 10, line 464 and 474).

  1. Typo in table 2: “Phage”

Author’s response: We are sincerely sorry for our typo and the correction has been made.

  1. Ensure all abbreviations used in the manuscript are fully accounted for in the list; several are currently missing.

Author’s response: Thank you for your suggestion and the complete list is revised accordingly.

Abbreviation list:

PD-1: programmed cell death protein 1

CTLA-4: cytotoxic T-lymphocyte-associated protein 4

ICIs: immune checkpoint inhibitors

TME: tumor microenvironment

RT: radiotherapy

TAAs: tumor-associated antigens

MRI: magnetic resonance imaging

ERK: extracellular signal-regulated kinase

MAPK: mitogen-activated protein kinase

NF-κB: nuclear factor kappa B

HNSCC: head and neck squamous cell carcinoma

ACT: adaptive cell therapy

HPV: human papillomavirus

FDA: US Food and Drug Administration

ORR: overall response rate

BOR: best overall response

Tregs: regulatory T cells

R/M: relapsed/metastatic

HNCs: head and neck cancers

MDSCs: myeloid-derived suppressor cells

tTregs: thymus-derived Tregs

pTregs: periphery-derived Tregs

FOXP3: Forkhead Box P3

OS: overall survival

PFS: progression-free survival

EFS: event-free survival

CI: confidence interval

TILs: tumor-infiltrating lymphocytes

HR: hazard ratio

BM: bone marrow

DCs: dendritic cells

CTLs: cytotoxic T lymphocytes

NK: natural killer

NKT: natural killer T cells

TAMs: tumor-associated macrophages

pMN: premetastatic niche

MMP9: matrix metalloprotease 9

MET: mesenchymal-epithelial transition

VEGF: vascular endothelial growth factor

GM-CSF: granulocyte-macrophage colony-stimulating factor

TNF- α: tumor necrosis factor α

TCRs: T-cell receptors

TIM-3: T cell immunoglobulin mucin-3

LAG-3: lymphocyte activation gene 3

VISTA: V domain Ig suppressor of T cell activation

TIGIT: T cell immunoglobulin and ITIM domain

PD-L1: programmed death ligand-1

PFS: progression-free survival

mAb: monoclonal antibody

MTD: maximum tolerated dose

CT: chemotherapy

IFN-γ: interferon-gamma IFN-γ

IMRT: intensity-modulated radiotherapy

Reviewer 2 Report

Comments and Suggestions for Authors

The authors of the present work described the landscape of Immunotherapeutic approaches for oral cancer therapy.  

The manuscript looks like well written and organized. The authors have presented an interesting topic in the field of oral cancer treatments. The paper should be considered after major revisions.

1.       In the abstract, it is reported that CTLA-4 inhibitors are pembrolizumab and nivolumab (lines 25-26);

2.       In the first part of the manuscript the authors should add a short overview of the use of chemotherapy, drugs combination, radiotherapy and surgery in the treatment of oral cancers;

3.       The topic of personalized medicine is mentioned only in the last paragraph of the manuscript. This therapeutic approach represents an important challenge of modern medicine and the authors should better described the last innovation in this field. The following references should be included in the manuscript: “Precision medicine in Head and Neck Cancers: Genomic and Preclinical approaches. Doi: 10.3390/jpm12060854” and “To Tip or Not to Tip: A New Combination for Precision Medicine in Head and Neck Cancer. doi: 10.1158/0008-5472.CAN-23-1858.”.

Author Response

Reviewer 2# The authors of the present work described the landscape of Immunotherapeutic approaches for oral cancer therapy. The manuscript looks like well written and organized. The authors have presented an interesting topic in the field of oral cancer treatments. The paper should be considered after major revisions.

  1. In the abstract, it is reported that CTLA-4 inhibitors are pembrolizumab and nivolumab (lines 25-26);

Author’s response: We are sincerely sorry for our careless mistake. The inhibitors are now in correct form as ipilimumab and tremelimumab (page 1, line 24).

  1. In the first part of the manuscript the authors should add a short overview of the use of chemotherapy, drugs combination, radiotherapy and surgery in the treatment of oral cancers;

Author’s response: Thank you very much for your valuable suggestion. In response to your suggestions, we have address the points in page 1-2, line 45-50. “Traditional treatment approaches such as surgery, chemotherapy, radiotherapy, and combination therapy are effective in the treatment of HNSCC. The surgical approach is an established gold standard for the initial treatment of HNSCC patients. However, around 94% (N= 219/234) of the patients had a recurrence within 18 months after surgery. Alternatively, radiotherapy and/or chemotherapy are often used to treat HNSCC. Although reducing tumors, those treatments also pose adverse effects [9-11].”

  1. The topic of personalized medicine is mentioned only in the last paragraph of the manuscript. This therapeutic approach represents an important challenge of modern medicine and the authors should better have described the last innovation in this field. The following references should be included in the manuscript: “Precision medicine in Head and Neck Cancers: Genomic and Preclinical approaches. Doi: 10.3390/jpm12060854” and “To Tip or Not to Tip: A New Combination for Precision Medicine in Head and Neck Cancer. doi: 10.1158/0008-5472.CAN-23-1858.”.

Author’s response: Thank you very much for your valuable suggestion. In response to your suggestion, we have added the references and revised our paper (page 13, line 492-501). “Two major approaches of precision medicine such as multi-OMICS and personalized preclinical platform are particularly considered for individualized treatment of HNSCC patients with improved efficacy. The first approach can assist in the identification of tumor behavior, employing biomarkers. Only a very small group of patients with R/M HNSCC can benefit from ICIs due to a lack of validated biomarkers. The selection of appropriate biomarkers may facilitate the achievement of clinically meaningful therapeutic feedback. The second approach could predict drug sensitivity in clinical samples. Preclinical cancer models such as patient-derived xenografts (PDX) have remarkable parallels at the genetic, transcriptomic, and proteomic levels that may strengthen biomarker-based clinical trials and improve the management of HNSCC [130, 131].”

Reviewer 3 Report

Comments and Suggestions for Authors

The submitted review article concentrates on immune checkpoint inhibitor therapy.

The Introduction is brief, and correctly starts the topic.

The review article is well-written, contains valuable information.

Some provocative comments:

1.       Since the actual approved Pembrolizumab therapy achieves response in 20% of HNSCC patients, as well as in 20% of colon cancer patients, meaning that for 80% of the patients it is ineffective. Moreover, the reasons for the success for the 20% are not clearly rational. This raises the question: is it useful to deal further with a 20% success therapy at all, or find predictive factors for the 20% success patients to match them with Pembrolizumab, or invest more research into follow up treatment for patients (80%!) who received unsuccessful ICI therapy? Several arguments against these points are present in the review article. Taking these doubting counterarguments and delivering achievements against these points will make this review a high assessed lively article.

2.       Page 3, line 102

“….They effectively inhibit the activation of TAA-specific effector T cells”. This is a cellular mechanisms-oriented journal. Could you give short clues how Tregs effectively inhibit the activation of TAA-specific effector T cells?

3.       Page 3. Line 108

Which ratio of CD8+/FOXP3+ and CD8+/CD4+ cells is suppressive and which is efficient in terms of the tumor killing function of immune system? Can this be stated or the literature delivers contradictive data?

This is especially interesting in the light of the later cited data that “high-level infiltration of Tregs was remarkably associated with shorter overall survival (OS) in most solid tumors, including melanoma, renal, cervical, and breast cancers, whereas opposite results were obtained in colorectal, esophageal, and head and neck tumors.”

4.       Page 4, lines 157-168. Are the immune checkpoint molecules in tumor tissue, or on tumor cells or on corresponding immune cells upregulated at the same time or differently? This means that, if we block one immune checkpoint pathway several ones are still present and active, which predestinates the clinical uselessness of the immune checkpoint blockade therapy. Could you argue against this provocative point or bring this discussion to the review article?

5.       Page 8, lines 384-385. “When RT is combined with immune checkpoint inhibitors (ICIs), it can potentiate the synergistic effects…”. Immune checkpoint inhibitors are given to HNSCC patients which were routinely treated before with routine standard therapy including Cisplatin and RT. These patients have RCT resistant tumor. How can this tumor benefit from the RT-ICI combination? In my knowledge, there is no approval for ICI as part of the first line standard treatment as RCT.

Author Response

Reviewer 3# The submitted review article concentrates on immune checkpoint inhibitor therapy. The Introduction is brief, and correctly starts the topic. The review article is well-written, contains valuable information.

  1. Since the actual approved Pembrolizumab therapy achieves response in 20% of HNSCC patients, as well as in 20% of colon cancer patients, meaning that for 80% of the patients it is ineffective. Moreover, the reasons for the success for the 20% are not clearly rational. This raises the question: is it useful to deal further with a 20% success therapy at all, or find predictive factors for the 20% success patients to match them with Pembrolizumab, or invest more research into follow up treatment for patients (80%!) who received unsuccessful ICI therapy? Several arguments against these points are present in the review article. Taking these doubting counterarguments and delivering achievements against these points will make this review a high assessed lively article.

Author’s response: Thank you for your valuable comment. In page 7, line 310-313 we addressed the achievement of Pembrolizumab as “In advanced HNSCC patients, treatments with Pembrolizumab have shown high survival rates (6-month OS rate: 58% and 12-month OS rate: 38%) and no adverse event-related deaths, an outcome rarely seen with existing cytotoxic or targeted therapies [104].”

In addition to bring counterarguments against the above-mentioned points we address the following sentence in our current version of manuscript in page 2, line 73-76…. “Although a large group of patients do not take advantage of the clinical advantages of ICI therapy, it is valuable to identify valid predictive biomarkers and advanced treatment approaches for most patients [22].”

  1. Page 3, line 102

“….They effectively inhibit the activation of TAA-specific effector T cells”. This is a cellular mechanisms-oriented journal. Could you give short clues how Tregs effectively inhibit the activation of TAA-specific effector T cells?

Author’s response: Thank you very much for your time and valuable comment. In response to your question, we have revised paragraph in page 3, line 101-106. “Tregs directly inactivate TAA-specific effector T cells by releasing a variety of immunosuppressive cytokines (IL-10, IL-35, TGF-β), attenuating the secretion of IL-2, which is essential for effector T cells survival, and ultimately inhibiting the immune responses against tumors. Alternatively, Tregs also express CTLA-4 which diminishes co-stimulatory signals to effector T cells by downregulating CD80/CD86 in dendritic cells. [33].”

  1. Page 3. Line 108

Which ratio of CD8+/FOXP3+ and CD8+/CD4+ cells is suppressive and which is efficient in terms of the tumor killing function of immune system? Can this be stated or the literature delivers contradictive data?

This is especially interesting in the light of the later cited data that “high-level infiltration of Tregs was remarkably associated with shorter overall survival (OS) in most solid tumors, including melanoma, renal, cervical, and breast cancers, whereas opposite results were obtained in colorectal, esophageal, and head and neck tumors.”

Author’s response: Thank you very much for your time and valuable comment. To address your question, we have included the following paragraph in our current version of manuscript in page 3, line 110-118. “The ratio between different subgroups of T cells thus provides an informative measure for tumor occurrence and progression, for example, the CD8+/CD4+ and CD8+/FOXP3+ ratios are the most-used measurement for the potency of anti-tumor immune activity [36]. Specifically, the above two subset ratios are only considered tumor suppressive if the CD8+ T cells are increased compared to FOXP3+ and/or CD4+ T cells after RT or ICI treatment. Between CD8+/CD4+ and CD8+/FOXP3+ ratios, the latter is more efficient in terms of antitumor immune activity. Considering that one of the most important functions of Tregs is to suppress other immune cells, the high CD8+/FOXP3+ ratio denotes an escape of tumor cells from the immune surveillance [37].”

  1. Page 4, lines 157-168.

Are the immune checkpoint molecules in tumor tissue, or on tumor cells or on corresponding immune cells upregulated at the same time or differently? This means that, if we block one immune checkpoint pathway several ones are still present and active, which predestinates the clinical uselessness of the immune checkpoint blockade therapy. Could you argue against this provocative point or bring this discussion to the review article?

Author’s response: Thank you for your valuable question. To address this question, we added the following paragraph in our current version of manuscript in page 4, line 174-176. “ICIs specifically target a single pathway while others might remain active, thus, it is important to consider combining other target therapies to achieve synergistic anti-tumor effects, such as multiple ICI therapies”.

  1. Page 8, lines 384-385.

“When RT is combined with immune checkpoint inhibitors (ICIs), it can potentiate the synergistic effects…”. Immune checkpoint inhibitors are given to HNSCC patients which were routinely treated before with routine standard therapy including Cisplatin and RT. These patients have RCT resistant tumor. How can this tumor benefit from the RT-ICI combination? In my knowledge, there is no approval for ICI as part of the first line standard treatment as RCT.

Author’s response: Thank you very much for your valuable comment. To address your comment, we added the following paragraph in the current version of our manuscript in page 9, line 432-437. “ICI treatment is intended for recurrent or metastatic HNSCC that has previously responded to RT, especially for those whose immune cells are being transformed into immunosuppressive and radio-resistant phenotypes. In a phase II study (NCT02641093), the employment of pembrolizumab as adjuvant RT has been shown to improve the survival of patients with locally advanced HNSCC [123].”

Reviewer 4 Report

Comments and Suggestions for Authors

Dear authors,
the article is well done and covers the different ways of treating an HNSCC.
There are a few questions to correct regarding the initial phase of the article.
-In the title you wrote ... oral carcinoma.
Afterwards, sometimes you mention oral carcinoma, sometimes HNSCC, sometimes HPV oropharyngeal carcinoma.
In my opinion, these parts of the article should be corrected because they create doubts and problems in the interpretation of the work.
-The etiopathogenesis of the tumor, it was written, is based on smoking, alcohol, other factors, and leukoplakia and erythema (I believe erythroplachia). Leukoplakia and erythroplachia are not etiopathogenetic agents that induce carcinoma, they are precancerous. this part for me should also be well explained and corrected.
You wrote in the abstract that pembrolizumab and nivolumab are CTLA-4 inhibitors. This is not true, but ipililumab and nivolumab are CTLA-4 inhibitors. This is not true, but ipililumab and nivolumab are CTLA-4 inhibitors. This is not true, but ipililumab is an etiologic agent that induces carcinoma. This is not true, but ipilimumab is a CTLA-4 inhibitor. In fact, in the text you wrote that the only study that used ipilimumab and nevolumab+pembrolizumab showed no therapeutic efficacy.

This part should be well written and well clarified.

Author Response

Reviewer 4# The article is well done and covers the different ways of treating an HNSCC. There are a few questions to correct regarding the initial phase of the article.
1. In the title you wrote ... oral carcinoma. Afterwards, sometimes you mention oral carcinoma, sometimes HNSCC, sometimes HPV oropharyngeal carcinoma.
In my opinion, these parts of the article should be corrected because they create doubts and problems in the interpretation of the work.

Author’s response: Thank you for your time and valuable suggestion. To reduce the confusion, we have revised our title to “The use of immune regulation in treating head and neck squamous cell carcinoma (HNSCC)” and tried to continuously use HNSCC”.

  1. The etiopathogenesis of the tumor, it was written, is based on smoking, alcohol, other factors, and leukoplakia and erythema (I believe erythroplachia). Leukoplakia and erythroplachia are not etiopathogenetic agents that induce carcinoma, they are precancerous. this part for me should also be well explained and corrected.

Author’s response: Thank you for your time and valuable suggestion. We are sincerely sorry for our careless mistake. The “leukoplakia and erythema” are now removed and the paragraph is revised as following (page 1, line 41-43). “HNSCC can be induced by many factors, including long-term alcoholism, poor oral hygiene, excessive sun exposure, betel nut chewing, and cigarette smoking [4-6].”

  1. You wrote in the abstract that pembrolizumab and nivolumab are CTLA-4 inhibitors. This is not true, but ipililumab and nivolumab are CTLA-4 inhibitors. This is not true, but ipililumab and nivolumab are CTLA-4 inhibitors. This is not true, but ipililumab is an etiologic agent that induces carcinoma. This is not true, but ipilimumab is a CTLA-4 inhibitor. In fact, in the text you wrote that the only study that used ipilimumab and nevolumab+pembrolizumab showed no therapeutic efficacy.

This part should be well written and well clarified

Author’s response: We apologize for our careless mistake. Thank you for your time and valuable suggestion. In our current version of manuscript, we correct our mistake in page 1, line 24. “The CTLA-4 inhibitors, such as ipilimumab and tremelimumab”.

To address your question, we have included the following paragraph in our current version of manuscript in page 8, line 372-378. “A randomized phase II clinical trial with 267 relapsed/ metastatic R/M HNSCC patients showed that the combination of durvalumab and tremelimumab resulted in clinically relevant overall survival and manageable toxic effects [115]. The clinical trial CheckMate 651 (NCT02823574) comprehensively demonstrated that a combination of nivolumab and ipilimumab is an excellent disease control agent in R/M HNSCC, increasing the median OS from 13.5 to 13.9 months compared to the EXTREME therapy [116].”

Reviewer 5 Report

Comments and Suggestions for Authors

 The topic of this review is of interest in the field of oral cancer, where new therapeutic approaches are much needed. However, issues affecting concepts, clarity, bibliography and English make me consider this work unsuitable for publication. I don't think it helps interested readers understand with clarity and rigor the advances that are taking place in this field.

 Below are some of the reasons that lead me to recommend the rejection of this article. It is not a list of recommendations, but a few examples on which I have based my decision.

 Some issues affecting concepts:

 Abstract Line 25_ “The CTLA-4 inhibitors, such as pembrolizumab and nivolumab,…”

Both are anti PD-1 antibodies, not CTLA-4 inhibitors.

 Line 45_ “Oral cancer can be caused by many factors, including long-term alcoholism, poor oral hygiene, excessive sun exposure, betel nut chewing, long-term foreign body stimulation, malnutrition, leukoplakia or erythema of the mucosa, and oral ulcers [6,7].”

The authors do not mention smoking as one of the main causes of oral cancer.

 Line 71_ “Therefore in 2017, nivolumab was approved as a second-line therapy for platinum-resistant R/M HNSCC…”

Nivolumab received FDA approval as a second-line therapy for platinum-resistant R/M HNSCC in November 2016 (https://drugs.com/history/opdivo.html). On 2017, the results of the Phase III Checkmate 141 were published in Lancet Oncology.

Line 158_”The term immune checkpoint (IC) refers to programmed death receptors and their ligands.”

Immune checkpoints are pathways regulating the immune system. There are many immune checkpoint molecules, including stimulatory and inhibitory, not only programmed death receptors and their ligands.

  Some examples of outdated or incorrect bibliography:

 Line 59_Recent work introduced molecular targeted therapy by immune checkpoint inhibitors (ICIs), which are becoming increasingly important [13].”

Reference 13 does not apply: Multinational, randomized study to compare radiotherapy alone with radiotherapy plus Cetuximab.

 Line 166_ “Other important ICs include T cell immunoglobulin mucin-3 (TIM-3), lymphocyte activation gene 3 (LAG -3), V domain Ig suppressor of T cell activation (VISTA), and others [57-59].”

Reference 59 does not apply: EXTL3 reported as an anti-oncogene. There is no evidence to consider it “Other important ICs”.

 Line 86_”During tumor initiation, development, and progression of oral cancer immune system plays a key role. It has been observed that immunosuppressed individuals are more likely to develop oral cancer and their prognosis is relatively poor [24].”

There are more updated references than 24 (2015). For example: Is Systemic Immunosuppression a Risk Factor for Oral Cancer? A Systematic Review and Meta-Analysis. Cancers (Basel) 2023 Jun 6;15(12):3077. doi: 10.3390/cancers15123077.

 Line 111_ “High-level infiltration of Tregs was remarkably associated with shorter overall survival (OS) in most solid tumors, including melanoma, renal, cervical, and breast cancers, whereas opposite results were obtained in colorectal, esophageal, and head and neck tumors [36].”

Reference 36 is a 2015 general systematic review (meta-analysis) covering different cancer types. Since then, there have been significant advances in the characterization of the immune phenotype in head and neck cancer not mentioned in this review. At least three more recent reports have explored the levels of CTLs and Tregs and their prognostic value specifically in head and neck cancer [Cancers 2021;13(4):781 and Oncoimmunology 2017;6:e1356148 meta-analysis; Cancers 2019;11(9):1398 ].

 Some unclear sentences:

 Line 174_”In HNSCC, PD-L1 is produced by abnormal PD-1 signaling, leading to tumor-associated immunosuppression [62].”

 Line 459_”In some cases, immune checkpoint inhibitors may induce low positive response and induce adverse effects when combined with chemotherapy, permitting more potent and secure treatment strategies.”

 Some poorly constructed sentences:

 Line 68_”In 2016, the US Food and Drug Administration (FDA) approved as immunotherapeutic are two monoclonal antibodies against PD-1, nivolumab and pembrolizumab, for the treatment of patients with recurrent-metastatic (R/M) HNSCC refractory to platinum-based therapy.”

 Line 137_”Given that TAMs are one of the most pivotal immune cells engaged in the foundation of an immunosuppressive TME, therefore, they play a crucial role in oral carcinogenesis by influencing tumor growth, progression, and metastasis.”

 Line 433_”Significant progress has been made in oral cancer immune therapy in recent years. In recent years, a novel pattern…”

 Line 438_”To further improve the efficacy of oral cancer treatment in the future, several innovative strategies such as personalized immunotherapy, combination therapy, dendritic cell vaccines, targeted therapy, and peptide therapy are under discussion to improve oral cancer therapy.”

 Line 443_”Recent cutting-edge techniques in proteomics, immune profiling confirm clinical personalities to create a systematic plan of immunotherapy for…”

 Line 463_”The ability to specifically target cancer cells makes peptide therapy more effective than traditional cancer treatments, making peptide therapy a promising candidate for…”

 Some minor errors:

 New paragraph should start at the beginning of Line 118 rather than line 124.

 The definition for Tumor-associated macrophages (TAMs) should appear when first mentioned: Line 127

 Table 1: “Phase” instead of “Phage”

Author Response

Reviewer 5# The topic of this review is of interest in the field of oral cancer, where new therapeutic approaches are much needed. However, issues affecting concepts, clarity, bibliography and English make me consider this work unsuitable for publication. I don't think it helps interested readers understand with clarity and rigor the advances that are taking place in this field.

Below are some of the reasons that lead me to recommend the rejection of this article. It is not a list of recommendations, but a few examples on which I have based my decision.

  1. Abstract Line 25_ “The CTLA-4 inhibitors, such as pembrolizumab and nivolumab,…” Both are anti PD-1 antibodies, not CTLA-4 inhibitors.

Author’s response: We apologize for our careless mistake. The inhibitors are in correct form as “The CTLA-4 inhibitors, such as ipilimumab and tremelimumab” (age 1, line 24).

  1. Line 45_ “Oral cancer can be caused by many factors, including long-term alcoholism, poor oral hygiene, excessive sun exposure, betel nut chewing, long-term foreign body stimulation, malnutrition, leukoplakia or erythema of the mucosa, and oral ulcers [6,7].” The authors do not mention smoking as one of the main causes of oral cancer.

Author’s response: We apologize for our careless mistake. In this revision, we have added the following as “HNSCC can be induced by many factors, including long-term alcoholism, poor oral hygiene, excessive sun exposure, betel nut chewing, and cigarette smoking [4-6]” (page 1, line 41-43).

  1. Line 71_ “Therefore in 2017, nivolumab was approved as a second-line therapy for platinum-resistant R/M HNSCC…”

Nivolumab received FDA approval as a second-line therapy for platinum-resistant R/M HNSCC in November 2016 (https://drugs.com/history/opdivo.html). On 2017, the results of the Phase III Checkmate 141 were published in Lancet Oncology.

Author’s response: We are sincerely sorry for our careless mistake. In this revision, we have added the following as “In 2016, the US Food and Drug Administration (FDA) approved two PD-1 monoclonal antibodies, nivolumab and pembrolizumab, for the treatment of platinum-resistant recurrent metastatic HNSCC. The affirmative results from the CheckMate 141 and KEYNOTE-048 trials validated that the PD-1 inhibitors improve survival and response in HNSCC patients compared to the single agent or single agent with chemotherapy group [19, 20]” (page 2, line 65-70).

  1. Line 158_”The term immune checkpoint (IC) refers to programmed death receptors and their ligands.”

Immune checkpoints are pathways regulating the immune system. There are many immune checkpoint molecules, including stimulatory and inhibitory, not only programmed death receptors and their ligands.

Author’s response: We are sincerely sorry for our careless mistake. In this revision, we have added the following as “Immune checkpoints are important in supporting the proper function of the immune system. Several immune checkpoint signaling pathways are especially essential in the tumor immune microenvironment, consisting of programmed death receptors and their ligands such as PD-L1 and PD-L2, as well as stimulatory (e.g. CD40L and CD70) and inhibitory (e.g. PD-1, CTLA-4, LAG-3, TIM-3) [59]” (page 4, line 165-169).

Some examples of outdated or incorrect bibliography:

  1. Line 59_ “Recent work introduced molecular targeted therapy by immune checkpoint inhibitors (ICIs), which are becoming increasingly important [13].”

Reference 13 does not apply: Multinational, randomized study to compare radiotherapy alone with radiotherapy plus Cetuximab.

Author’s response: We apologize for the mistake. We have removed the original reference [13] and cited the following correct one in the revision. “Recent work introduced molecular targeted therapy by immune checkpoint inhibitors (ICIs), which are becoming increasingly important [14].”

Kitamura, N., et al., Current Trends and Future Prospects of Molecular Targeted Therapy in Head and Neck Squamous Cell Carcinoma. Int J Mol Sci, 2020. 22(1)

  1. Line 166_ “Other important ICs include T cell immunoglobulin mucin-3 (TIM-3), lymphocyte activation gene 3 (LAG -3), V domain Ig suppressor of T cell activation (VISTA), and others [57-59].”

Reference 59 does not apply: EXTL3 reported as an anti-oncogene. There is no evidence to consider it “Other important ICs”.

Author’s response: We apologize the mistake and now have removed the original reference [59] and cite the correct one with updated information. “Other important ICIs include T cell immunoglobulin mucin-3 (TIM-3), lymphocyte activation gene 3 (LAG -3), V domain Ig suppressor of T cell activation (VISTA), and TIGIT (T cell immunoglobulin and ITIM domain) and Siglec-15 have shown their applications in advanced malignancy such as melanoma, NSCLC, and HNSCC [62-64]” (page 4, line 179-182).

The newly added reference (ref. 64)- Li, B., et al., Immune Checkpoint Inhibitors Combined with Targeted Therapy: The Recent Advances and Future Potentials. Cancers (Basel), 2023. 15(10).

  1. Line 86_”During tumor initiation, development, and progression of oral cancer immune system plays a key role. It has been observed that immunosuppressed individuals are more likely to develop oral cancer and their prognosis is relatively poor [24].”

There are more updated references than 24 (2015). For example: Is Systemic Immunosuppression a Risk Factor for Oral Cancer? A Systematic Review and Meta-Analysis. Cancers (Basel) 2023 Jun 6;15(12):3077. doi: 10.3390/cancers15123077.

Author’s response: Thank you very much for your time and suggestion. In our current version of manuscript, we added the following sentence in page 2, line 89-90 with your recommended reference….“Immunosuppression has been reported to be one of the main risk factors (0.2% to 1%) for the development of HNSCC and the prognosis is relatively poor [26]”.

  1. Line 111_ “High-level infiltration of Tregs was remarkably associated with shorter overall survival (OS) in most solid tumors, including melanoma, renal, cervical, and breast cancers, whereas opposite results were obtained in colorectal, esophageal, and head and neck tumors [36].”

Reference 36 is a 2015 general systematic review (meta-analysis) covering different cancer types. Since then, there have been significant advances in the characterization of the immune phenotype in head and neck cancer not mentioned in this review. At least three more recent reports have explored the levels of CTLs and Tregs and their prognostic value specifically in head and neck cancer [Cancers 2021;13(4):781 and Oncoimmunology 2017;6:e1356148 meta-analysis; Cancers 2019;11(9):1398 ].

Author’s response: Thank you very much for your time and suggestion. According to your suggestion in our current version of manuscript, we added the following sentence in page 3, line 123-128 with 3 more recently reported studies. “Recent reports demonstrated that FOXP3+ tumor-infiltrating lymphocytes (TILs) were associated with improved overall survival (OS) in HNSCC patients (HR: 0.80; 95%CI: 0.70–0.92) [40, 41]. Another study showed that both a high proportion of CD4+ (HR: 0.77; 95% CI: 0.65–0.93) and a high proportion of CD8+ TILs (HR: 0.64; 95% CI: 0.47–0.88) significantly reduced the risk of death and improved overall survival in HNSCC patients [42]”.

Some unclear sentences:

  1. Line 174_”In HNSCC, PD-L1 is produced by abnormal PD-1 signaling, leading to tumor-associated immunosuppression [62].”

Author’s response: In this version, the paragraph is now read as “The interaction between PD-1 and PD-L1 is demonstrated to attenuate T cell receptor-mediated lymphocyte proliferation and cytokine secretion such as IFNs, TNF-α, and VEGF [67]. Moreover, Tregs facilitate PD-1 to engage with PD-L1 and suppress CD4+ and CD8+ T effector cells in the TME [68]” (page 4, line 192-195).

  1.  Line 459_”In some cases, immune checkpoint inhibitors may induce low positive response and induce adverse effects when combined with chemotherapy, permitting more potent and secure treatment strategies.”

Author’s response: In this version, the paragraph is now read as “In some cases, immune checkpoint inhibitors may induce limited efficacy and adverse effects with CT and/or RT, thus other safe and effective treatment approaches should be considered.” (page 13-14, line 517-519).

Some poorly constructed sentences:

  1. Line 68_”In 2016, the US Food and Drug Administration (FDA) approved as immunotherapeutic are two monoclonal antibodies against PD-1, nivolumab and pembrolizumab, for the treatment of patients with recurrent-metastatic (R/M) HNSCC refractory to platinum-based therapy.”

Author’s response: In this version, the paragraph is now read as“ In 2016, the US Food and Drug Administration (FDA) approved two PD-1 monoclonal antibodies, nivolumab and pembrolizumab, for the treatment of platinum-resistant recurrent metastatic HNSCC.” (page 2, line 65-66).

  1. Line 137_”Given that TAMs are one of the most pivotal immune cells engaged in the foundation of an immunosuppressive TME, therefore, they play a crucial role in oral carcinogenesis by influencing tumor growth, progression, and metastasis.”

Author’s response: In this version, the paragraph is now read as “TAMs play a crucial role in oral carcinogenesis by influencing tumor growth, progression, and metastasis through immunosuppression. There are two types of TAMs, the classically activated M1 phenotype with anti-tumor activity, and the alternatively activated M2 phenotype, that possess tumor-promoting activity [50].” (page 3, line 145-148).

  1. Line 433_”Significant progress has been made in oral cancer immune therapy in recent years In recent years, a novel pattern…..

Author’s response: In this version, the paragraph is now read as..“Immunotherapy has emerged as one of the most advanced methods in cancer treatment today. Although remarkable progress has been made in the immunotherapy of HNSCC, a newly identified pattern of cancer progression, known as hyperprogressive disease (HPD), in which patients rapidly deteriorate in the early stages of treatment. Therefore, further studies are needed to comprehensively explore other possible therapeutic strategies in HNSCC [129]” (page 13, line 482-487).

  1. Line 438_”To further improve the efficacy of oral cancer treatment in the future, several innovative strategies such as personalized immunotherapy, combination therapy, dendritic cell vaccines, targeted therapy, and peptide therapy are under discussion to improve oral cancer therapy.”

Author’s response: In this version, the paragraph is now read as.“To improve and optimize the efficacy and safety of future HNSCC therapies, several advanced strategies are currently being investigated, for example, combination therapy, dendritic cell vaccines, targeted therapy, peptide therapy, and precision immunotherapy.” (page 13, line 489-492)

  1. Line 443_”Recent cutting-edge techniques in proteomics, immune profiling confirm clinical personalities to create a systematic plan of immunotherapy for…”

Author’s response: In this version, the paragraph is now read as“ Current proteomics techniques can be employed to assess immune profiling in HNSCC patients, providing insights into new biomarkers and mutation sites. The information of individual immune profiling may also provide a personalized treatment plan to achieve effective health outcomes.” (page 13, line 504-508)

  1. Line 463_”The ability to specifically target cancer cells makes peptide therapy more effective than traditional cancer treatments, making peptide therapy a promising candidate for cancer management”

Author’s response:  In this version, the paragraph is now read as “Peptides are bioactive molecules with distinct properties, such as lower affinity and a shorter half-life in the body compared to antibodies, thereby enabling them to selectively bind to certain surface proteins of cancer cells, subsequently blocking their biological functions [136]. In addition, peptides in combination with immune checkpoint inhibitors may increase the specificity and efficacy of cancer immunotherapy by targeting specific tumor antigens [137].” (page 14, line 519-524)

Some minor errors:

  1. New paragraph should start at the beginning of Line 118 rather than line 124.

Author’s response: Thank you for your suggestion. The manuscript is now read as “Myeloid-derived suppressor cells (MDSCs) are a group of diverse cells that arise from the bone marrow (BM) and undergo numerous stages of differentiation until they evolve into macrophages, neutrophils, and dendritic cells (DCs) [43]. However, MDSCs are unnaturally generated, activated, and differentiated under pathological conditions, such as cancer” (page 3, line 129).

  1. The definition for Tumor-associated macrophages (TAMs) should appear when first mentioned: Line 127

Author’s response: Thank you for your suggestion. According to your suggestion we make correction in our current version of manuscript in page 3, line136… “Tumor-associated macrophages (TAMs)”.

  1. Table 1: “Phase” instead of “Phage”

Author’s response: We are sincerely sorry for our typo. In our current version of manuscript, we correct it. Page 11, in table- “Phase”.

Round 2

Reviewer 1 Report

Comments and Suggestions for Authors

I consider that the manuscript has improved significantly and has increased in quality, with everything being much clearer and more rigorous. The authors have addressed all the issues that I asked for and have revised the manuscript, so I consider it acceptable for publication.

Author Response

Thank you very much!!

Reviewer 2 Report

Comments and Suggestions for Authors

I consider the manuscript in the present form acceptable for the publication on Cells journal.

Author Response

Thank you!

Reviewer 5 Report

Comments and Suggestions for Authors

In my initial review of the work presented by Wang et al. I stated the reasons why I did not consider the article acceptable for publication in Cells. The list I gave was not the problems that the authors needed to correct, but simply some examples of the reasons that led me to make the decision.

Despite the changes made, the article still does not provide a clear picture of where the use of immunotherapy in oral cancer is heading. This is not the review I would suggest readers interested in delving deeper into this field to read.

Author Response

Thank you for the valuable suggestions. We will keep pushing ourselves to do better.